# Planting Date Effects on Cotton Lint Yield and Fiber Quality in the U.S. Southern High Plains

**Steven Mauget [1,*], Mauricio Ulloa [2] and Jane Dever [3]**

1 U.S. Department of Agriculture-Agricultural Research Service, Cropping Systems Research Laboratory, Wind Erosion and Water Conservation Research Unit, Lubbock, TX 79415, USA

2 U.S. Department of Agriculture-Agricultural Research Service, Cropping Systems Research Laboratory, Plant Stress and Germplasm Development Research Group, Lubbock, TX 79415, USA; Mauricio.Ulloa@ars.usda.gov

3 Department of Soil and Crop Sciences, Texas A & M AgriLife Research and Extension Center, Lubbock, TX 79403-6603, USA; jdever@ag.tamu.edu

* Correspondence: Steven.Mauget@ars.usda.gov; Tel.: +1-806-723-5237

**Abstract:** Cotton planting date effects in the U.S. Southern High Plains (SHP) were evaluated based on 11 years of May-planted and June-planted irrigated variety trials. Multiple cultivars planted in each year's trial allowed for the calculation of 153 yield effects and 162 effects in 5 fiber quality parameters. Yield and quality effects were considered in the context of related changes in total growing season degree days ($GDD_S$) and total cool hours (CHRS) during a boll formation period 80 to 110 days after planting. May planting increased $GDD_S$ and significantly increased yields in 8 of 10 years that comparisons could be made. Micronaire and fiber elongation were the most sensitive quality parameters to planting date. June planting resulted in increased CHRS every year and a significantly higher incidence of low micronaire in 7 of 11 years. In 7 of 11 years May planting significantly reduced fiber elongation relative to June planting. Analysis of SHP temperature data show that late-April to early-May planting dates may increase yield and micronaire by maximizing $GDD_S$ and minimizing CHRS. Although this practice may be optimal to the SHP environment it may also require high-vigor seed and pre-planting irrigation. Adapting genetics to an early planting strategy might include selecting for improved seed vigor and cold germination with acceptable yield and fiber quality traits.

**Keywords:** cotton production; lint yield; fiber quality; micronaire; cooling hours; growing degree days; genetics X environment X management

## 1. Introduction

Although the Southern High Plains (SHP) of west Texas are the leading cotton production region in the United States' leading cotton producing state, the region's high altitude presents challenges to cotton growth. Most of the area's top-producing counties are at elevations above 945 m (3100 ft), which leads to summer growing degree day (GDD) accumulations that usually fall short of levels that optimize crop development and yield. Peng et al. [1] estimated that SHP cotton yields under adequately-watered conditions might be maximized with a summer GDD accumulation of 1450.0 °C (2610.0 °F). However, under current SHP climate conditions, there is less than a 5% probability that 15 May–15 October GDD totals would exceed 1400.0 °C [2]. The SHP summer environment has been generally noted to be cool and degree-day limited relative to the requirements of cotton crop development [1,3,4].

To estimate the effects of elevation on SHP cotton yields, Mauget et al. [2] (hereafter M17) conducted DSSAT CROPGRO-Cotton simulations with daily West Texas Mesonet (WTM) weather

data inputs from higher elevation SHP stations and stations over the lower elevation Rolling Plains region to the east. These un-irrigated 'rainfed' simulations showed that as planting dates were delayed from early to late May, SHP lint yields at the higher elevation decreased while yields simulated under the warmer Rolling Plains conditions remained stable. This varying planting date effect with elevation was most obvious under higher initial soil moisture conditions. As cotton lint yield effects associated with GDD accumulation are most evident under limited water stress [1,5], the higher elevation yield effect indicates that lower GDD accumulations that result from later SHP planting dates can lead to lower lint yields. Mauget et al.'s [6] (hereafter M19) in silico yield, profit, and risk analyses showed that median rainfed SHP lint yields decrease, and the risk of economic loss increases, as planting is delayed from mid-March to early-June.

In addition to potential yield effects, cool summer SHP growing conditions might also affect fiber quality and value. High quality cotton fiber is uniformly strong, fine, long, and white, and is easily ginned and processed. Together with color grade and harvest trash content, lint value is a function of fiber strength, length, and uniformity, with lower values resulting in discounted value and higher values in premiums. Fiber micronaire, which measures the air permeability of compressed cotton fiber, is used as an indicator of fiber maturity and surface area. Lint with mid-range micronaire (3.7 to 4.2) is assigned a premium value, while lint with values less than 3.5 or greater than 4.9 are discounted. Lint with micronaire not within those ranges is assigned a base value. Past research has shown that low temperatures during the plant's boll formation period can play an important role in determining cotton fiber characteristics. Hessler et al.'s [7] field experiments showed that increased accumulations of cool hours (CHRS) below 21.1 °C (70 °F) during boll development were associated with decreased cellulose synthesis and sugar accumulation in the bolls. Wanjura and Barker [8] demonstrated that fiber micronaire and length were positively correlated with temperature during the boll formation period in field experiments. Lokhande and Reddy's [9] growth chamber experiments showed that increased temperature produced a quadratic increase in fiber micronaire and uniformity up to 26.0 °C, but decreased in both variables at higher temperatures. Fiber strength increased linearly with temperature, while fiber length responded similarly to micronaire and uniformity, with length increasing as temperatures rose in the 18.0–22.0 °C range, and decreasing above 22.0 °C. In these experiments micronaire was found to be the most responsive to temperature, with cool temperatures during the latter stages of boll development associated with values in the low-discount range (<3.5). Their literature review, and that of Liakatas et al. [10], summarizes past research highlighting the effects of low night temperatures during boll development on yield and fiber quality. Bradow and Davidonis' [11] review of cotton quality and processing research notes that the effects of temperature on micronaire have been reported in planting-date and flowering-date studies, with delayed planting dates leading to reduced micronaire in field experiments in South Africa [12] and the U.S. [13].

The SHP region's cool summer growing conditions can have real effects on fiber quality and profitability. In 2017 wet and cool conditions during the cotton crop's boll development period led to an immature crop with harvesting and quality issues [14,15]. In early November 2017, 47% of the bales tested in the U.S. Department of Agriculture's (USDAs) Lubbock classing office fell in the micronaire discount category below 3.5. This discount reduced the USDA loan program lint value by a minimum of \$0.044 kg$^{-1}$ (\$0.02 lb$^{-1}$), with potentially larger spot market discounts [16]. The region's short summer growing season may have even greater effects on yield-related profits. The M19 [6] rainfed crop simulations estimated an 85.3% probability that planting SHP cotton in mid-May rather than early June will increase lint yield, with a median increase of 163.7 kg ha$^{-1}$. This positive yield effect is 33.4% of the median rainfed yield reported by producers in the National Agricultural Statistics Service (NASS) District 11 SHP region during 2012–2016 (490.6 kg ha$^{-1}$), and 42.7% of the median yield reported in the neighboring District 12 (383.0 kg ha$^{-1}$).

Past work here [2,6] has relied on crop models to estimate the effects of environment and management on SHP cotton yield, but current cotton models lack the ability to simulate fiber quality effects [17]. However, planting date effects on fiber quality might be estimated from field experiments.

Fortunately, such experiments have been conducted in the SHP region by Texas A & M's AgriLife Research and Extension Centers [15,18–27]. During each year of 2007–2017 irrigated uniform cotton variety tests were carried out at the Lubbock, Halfway, and Lamesa AgriLife locations. At Lubbock these field trials were repeated with both May and June ('late-planted') planting dates during each of those 11 years. The Lubbock irrigated trials were conducted with a variety of cultivars that varied from year to year and were planted on dates separated by between 20 to 41 days (Table 1). In addition to lint yield, fiber quality data was recorded for each cultivar and planting date. As each year's trials were conducted with different planting dates but with the same cultivars under similar soil and available water conditions, they provide the opportunity to estimate planting date and temperature effects on fiber quality in a semi-controlled field setting. The effects of May and June planting dates on lint yields can also be compared with the simulated SHP planting date effects found in M19.

In agricultural research there is a growing recognition that producing crops successfully in changing or marginal environments may require an understanding of the interactions between genetics, environment, and management [28,29]. Based on this understanding, management and genetics might be adapted to a region's environment. Given the leading role of the SHP in U.S. cotton production there is a need to explore and define management approaches that are tailored to the region's cool growing conditions. To this end, the main goal here is to identify lint yield and quality effects associated with varying planting dates and temperature conditions in a field setting. Based on those production effects and an analysis of current SHP climate conditions during the cotton planting and growing season, we suggest an early planting strategy to optimize management to the region's environment. This strategy—together with cultivars adapted to cool planting conditions and shorter growing seasons—may increase yields via increased summer GDD accumulation. In addition, early planting may lower the risk of low-discounted fiber micronaire by limiting a crop's exposure to cool nighttime conditions during cotton's boll formation period.

In the following, Section 2 will describe WTM air and soil temperature data used to define 2005–2017 SHP climate conditions. That section will also describe the Lubbock AgriLife (hereafter 'AgriLife') variety test data and briefly describe statistical methods used to detect planting date effects on yield and fiber quality. Section 3 will describe the effect of planting date on yield and fiber quality, and features of the current SHP soil and air temperature climate that affect cotton planting, GDD accumulation, and boll development. Section 4 will discuss implications for an early planting strategy and options for tailoring cotton cultivar characteristics to a cooler SHP planting environment.

**Table 1.** Planting and harvest dates, total growing season growing degree days (GDD$_S$), and boll formation period cooling hours (CHRS) for May and June planting dates in the 2007–2017 Lubbock AgriLife variety trials. May–June Δ values indicate June values subtracted from the same year's May values.

| | May Planting | | | | June Planting | | | | May–June | | |
|---|---|---|---|---|---|---|---|---|---|---|---|
| Year | Planting Date (pdoy) | Harvest Date (hdoy) | GDD$_S$ (°C) | CHRS | Planting Date (pdoy) | Harvest Date (hdoy) | GDD$_S$ (°C) | CHRS | Δpdoy | ΔGDD$_S$ | ΔCHRS |
| 2007 | May 5 (125) | November 20 (324) | 1128.2 | 229.43 | June 15 (166) | November 20 (324) | 964.2 | 392.76 | −41 | 164.0 | −163.33 |
| 2008 | May 17 (138) | November 21 (326) | 1115.6 | 339.41 | June 10 (162) | November 21 (326) | 903.3 | 499.04 | −24 | 212.3 | −159.63 |
| 2009 | May 12 (132) | November 5 (309) | 1149.0 | 228.24 | June 10 (161) | November 12 (316) | 986.7 | 479.08 | −29 | 162.3 | −250.84 |
| 2010 | May 21 (141) | November 2 (306) | 1202.6 | 252.57 | June 10 (161) | November 22 (326) | 1026.0 | 354.56 | −20 | 176.6 | −101.99 |
| 2011 | May 5 (125) | November 10 (314) | 1587.9 | 45.59 | June 2 (153) | November 10 (314) | 1411.3 | 290.33 | −28 | 176.6 | −244.74 |
| 2012 | May 7 (128) | November 27 (332) | 1351.8 | 159.51 | June 11 (163) | November 27 (332) | 1116.6 | 426.83 | −35 | 235.1 | −267.32 |
| 2013 | May 21 (141) | December 12 (346) | 1238.5 | 277.16 | June 11 (162) | December 13 (314) | 1063.5 | 343.84 | −21 | 175.0 | −66.68 |
| 2014 | May 16 (136) | December 17 (351) | 1163.9 | 196.52 | June 13 (164) | December 17 (351) | 979.2 | 461.76 | −28 | 184.7 | −265.24 |
| 2015 | May 27 (147) | November 9 (313) | 1173.0 | 282.85 | June 18 (169) | November 12 (316) | 1023.9 | 374.84 | −22 | 149.1 | −91.99 |
| 2016 | May 9 (130) | October 27 (301) | 1284.7 | 246.84 | June 18 (170) | November 14 (319) | 1064.3 | 427.17 | −40 | 220.4 | −180.33 |
| 2017 | May 18 (138) | November 4 (308) | 1146.9 | 328.81 | June 12 (163) | November 27 (331) | 1012.6 | 440.09 | −25 | 134.3 | −111.28 |

## 2. Data and Methods

### 2.1. West Texas Mesonet Data

The temperature effects of planting date on cotton lint yield and fiber quality in the AgriLife variety tests were evaluated based on the related changes in two variables: Growing degree days accumulated between planting and harvest, and exposure to temperature conditions below 21.1 °C during the crop's boll formation period. The air and soil temperature data used here is derived from WTM weather stations maintained by Texas Tech University's National Wind Institute (Figure 1). In addition to other meteorological variables, these stations report air temperature at 2.0 m every 5 min. Soil temperature at 10 cm, which is used in Section 4 to calculate planting date conditions, is reported at 15 min intervals. Quality control procedures applied to archived WTM data during January 2005 to December 2017 are described in Schroeder et al. [30]. Additional tests applied to the hourly and daily values calculated here from the WTM sub-hourly temperature data are described in M17.

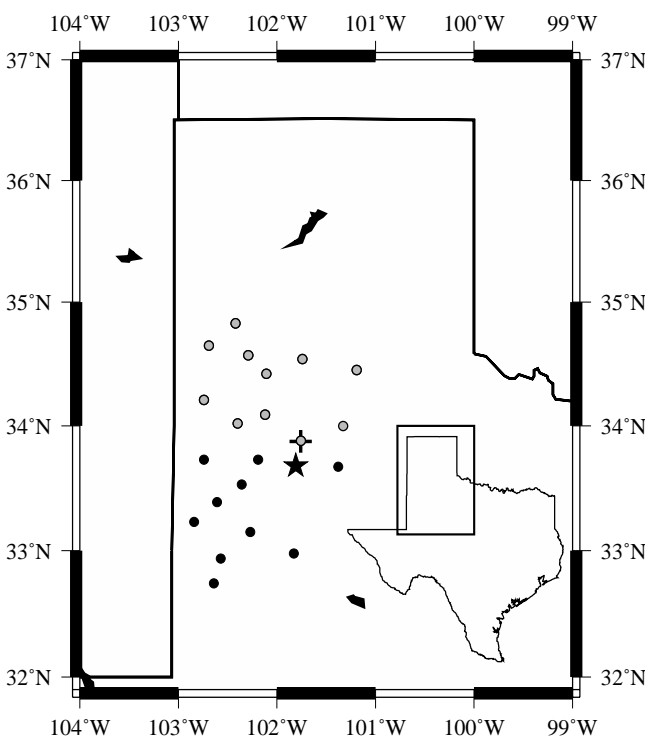

**Figure 1.** Locations of the 21 West Texas Mesonet (WTM) weather stations providing air and soil temperature data during 2005–2017. Gray (black) locations mark the 11 (10) northernmost (southernmost) stations. Crossed gray token marks the location of the Abernathy WTM station. Star marks the location of the Texas A & M Lubbock AgriLife research location.

As the Lubbock AgriLife research center does not maintain sub-daily temperature records needed to calculate cooling hours, the site's temperature conditions during each growing season of 2007–2017 were estimated from the sub-hourly data of the closest WTM station. This station is located 9.33 km (5.8 miles) NE of the city of Abernathy, and 21.7 km (13.5 miles) NNE of the Lubbock AgriLife location. Summer growing season cotton GDD ($GDD_S$) was calculated as the accumulation of daily average temperature ($T_d$) above a 15.6 °C (60 °F) threshold. These totals were summed between the planting day of year (pdoy) and harvest day of year (hdoy) of each year's May-planted and June-planted AgriLife variety tests, i.e.,

$$GDD_S = \sum_{d=pdoy}^{hdoy} Max(T_d - 15.6,\ 0.0) \tag{1}$$

Daily average temperature was calculated as the mean of the each day's maximum and minimum temperatures in the 5 min Abernathy data, i.e., $T_d = 0.5 \times (T_{Min} + T_{Max})$. During 2007–2017 May-planted Abernathy $GDD_S$ totals varied between 1115.6 °C and 1587.9 °C, while June-planted totals varied between 903.3 °C and 1411.3 °C (Table 1).

Based on Ritchie et al.'s [31] generic cotton phenology, a crops boll formation period was considered to occur 80 to 110 days after planting. As low temperature thresholds in the 20.0 to 22.0 °C range are widely cited in studies of the effects on cellulose synthesis and fiber properties [7,10,11,32,33], the 21.1 °C threshold used by Hessler et al. [7] was used to define cool hours. Thus for each May and June planting date, each growing season's total cool hours was defined as the sum of each day's total hours with temperature below 21.1 °C ($CHRS_d$) during the period 80 to 110 days after planting.

$$CHRS = \sum_{d=pdoy+80}^{pdoy+110} CHRS_d \qquad (2)$$

Abernathy May-planted CHRS totals varied between 45.6 and 339.4 h during 2007–2017, while June-planted totals varied between 290.33 and 499.04 h (Table 1). Relative to May planting dates, June planting dates reduced $GDD_S$ totals and increased a crops exposure to cooling hours (Figure 2). In addition to the Abernathy station data, air and soil temperature data from Figure 1's other 20 WTM stations was used in Section 3.3 to form regionally representative GDD and CHRS distributions, and to calculate variation in planting date conditions during 2005–2017.

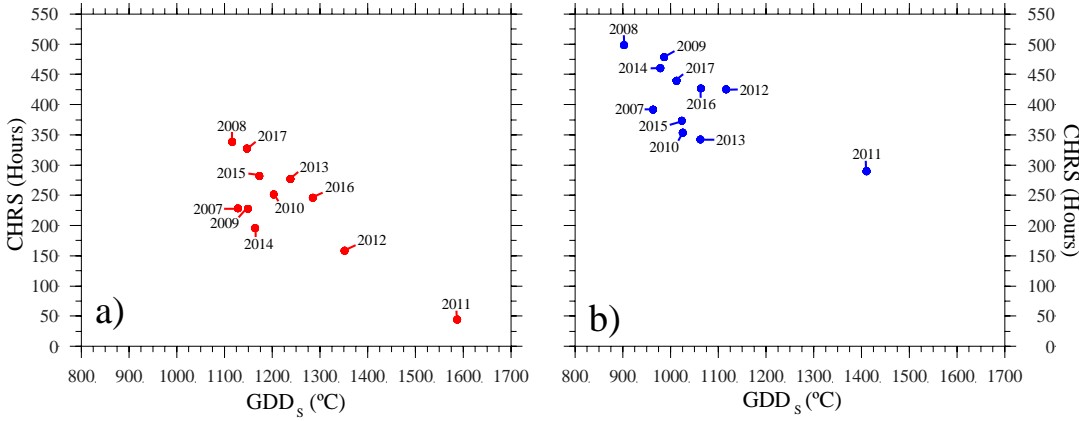

**Figure 2.** (**a**) Scatterplot of May-planted $GDD_S$ (X-axis) vs. May-planted CHRS (Y-axis) derived from the Abernathy mesonet station data during 2007–2017. (**b**) As in (a) for June-planted $GDD_S$ vs. June-planted CHRS.

## 2.2. AgriLife Yield and Fiber Quality Data

The AgriLife research center is located north of Lubbock, Texas at 33°41′23″ N, 101°49′42″ W. During each year of 2007–2017 between 26 to 55 cultivars were planted in the May variety tests, with fewer cultivars planted in June (Table 2). Among these cultivars, between 10 and 24 were planted each year on both of Table 1's May and June planting dates, allowing for yield and fiber quality comparisons of planting date effects for each cultivar. In 2014, early season cool temperatures led to varied yield effects among the June-planted entries [25]. As a result, no late-planted yields were reported in that year, and no comparisons of May-planted versus June-planted yields could be made. However, 2014 fiber quality data was reported, making it possible to make 153 yield and 162 fiber quality comparisons over the 11 years of field trials. The fiber quality effects evaluated here are derived from the reported data on micronaire (M), length (L), strength (S), uniformity (U), and elongation (E). Fiber analysis was conducted via high volume instrument (HVI) testing (Uster Technologies 1000 system) at Texas Tech University's Fiber and Biopolymer Research Institute.

**Table 2.** Total number of cultivars planted on Table 1's May and June planting dates. Common counts indicate the number of possible yield and fiber quality comparisons during each year. 'NA' indicates years when data was not available.

| | 2007 | 2008 | 2009 | 2010 | 2011 | 2012 | 2013 | 2014 | 2015 | 2016 | 2017 | Total |
|---|---|---|---|---|---|---|---|---|---|---|---|---|
| May-Planted Cultivars | 50 | 52 | 50 | 55 | 44 | 40 | 35 | 26 | 48 | 43 | 48 | 491 |
| June-Planted Cultivars | 27 | 20 | 15 | 20 | 25 | 20 | 25 | 16 | 28 | 15 | 28 | 239 |
| Common Yields | 15 | 13 | 14 | 15 | 18 | 13 | 14 | NA | 16 | 11 | 24 | 153 |
| Common Fiber Quality | 15 | 13 | 14 | 15 | 18 | 13 | 13 | 10 | 16 | 11 | 24 | 162 |

The 2007–2017 irrigated variety trials were conducted in a randomized complete block design with four replications, in 2-row plots with 9.15–12.2 m. (30–40 ft) long rows separated by 101.6 cm (40 in). These trials were planted in either an Acuff Loam (fine-loamy, mixed, superactive, thermic aridic Paleustoll) or Olton Clay Loam (fine, mixed, superactive, thermic aridic Paleustolls) soil. Additional production information for each year's trials can be found in Gannaway et al. [18] and Dever et al. [15,19–27]. Both May-planted and June-planted trials were irrigated during most years at similar levels (Table 3). In years where irrigation differed substantially, e.g., 2010, 2015, and 2016, in-season rainfall produced smaller relative differences in the total water available to the May-planted and June-planted crops. The yield and fiber quality effects calculated here are based on field trials planted on different planting dates, but with identical cultivars planted under generally similar soil type, irrigation, and precipitation conditions. As a result, these comparisons are controlled for cultivar and approximately controlled for soil type and total crop water. Although other factors may influence yield or quality, e.g., insect pressure, hail, chill, or blowing sand events between the May and June planting dates that might retard the development of the May-planted crop, these effects are assumed to be mainly attributable to the crop's different exposure to each growing season's GDDS and CHRS conditions.

**Table 3.** Total irrigation (cm) applied to May and June-planted variety trials during each year, and total in-season rainfall at the Lubbock AgriLife research site.

| | 2007 | 2008 | 2009 | 2010 | 2011 | 2012 | 2013 | 2014 | 2015 | 2016 | 2017 |
|---|---|---|---|---|---|---|---|---|---|---|---|
| May-Planted Irrigation | 14.7 | 36.3 | 27.7 | 21.1 | 65.3 | 41.4 | 40.6 | 21.8 | 13.7 | 9.4 | 8.9 |
| June-Planted Irrigation | 10.9 | 36.3 | 30.2 | 9.9 | 63.8 | 35.6 | 34.5 | 21.8 | 6.1 | 4.3 | 6.9 |
| In-Season Rainfall | NA | NA | 19.8 | 48.8 | 9.9 | 26.3 | 29.7 | 41.4 | 69.1 | 43.9 | 46.5 |

*2.3. Statistical Methods*

Planting date-related yield and fiber quality effects in the AgriLife data were identified and significance tested using R (RStudio version 1.1.383) statistical software [34]. The R code used to evaluate yield (yld_effect.R) and quality (fqual_effect.R) effects are included in the supplementary material, as are associated input data spread sheets containing the AgriLife yield (supplementary file 1: LBKYield2007–2017.xlsx) and fiber quality (supplementary file 2: LBKFiberQual2007–2017.xlsx) data. The statistical tests used are one sample and two-sided *t*-tests to determine whether a year's mean yield and fiber quality effects differ from 0.0 at a 95.0% ($p < 0.05$) or higher confidence level.

## 3. Results

*3.1. Planting Date Effects on Lint Yield*

Figure 3a plots the distribution of May-planted lint yields ($Y_M$) during 2007–2017, while Figure 3b scatterplots those yields on a year-by-year basis. Figure 3c,d show similar figures for the 153 June-planted yields ($Y_J$) of 2007–2013 and 2015–2017. Figure 3e shows the count distribution of 153 yield effects ($\Delta Y = Y_M - Y_J$) calculated by subtracting each cultivar's June yield in Figure 3d from the same year's and cultivar's May yield in Figure 3b. The Figure 3e bar plot marks the 25th (Q1), 50th (Q2), and 75th (Q3) percentiles of the $\Delta Y$ distribution. Figure 3f's scatterplots shows the $\Delta Y$

effects by year, and each year's mean yield effect. As in Figure 2 May data is plotted in red and June data in blue. May minus June data effects in Figure 3e,f and subsequent figures are plotted in green. Figure 3f's *t*-tests for each year's mean yield effect show that all of the associated population means differ from 0.0 at a 99.0% or higher confidence level ($p < 0.01$). Two years (2015, 2016) show a negative mean yield effect, while in eight years the earlier May planting dates resulted in significantly increased mean lint yield.

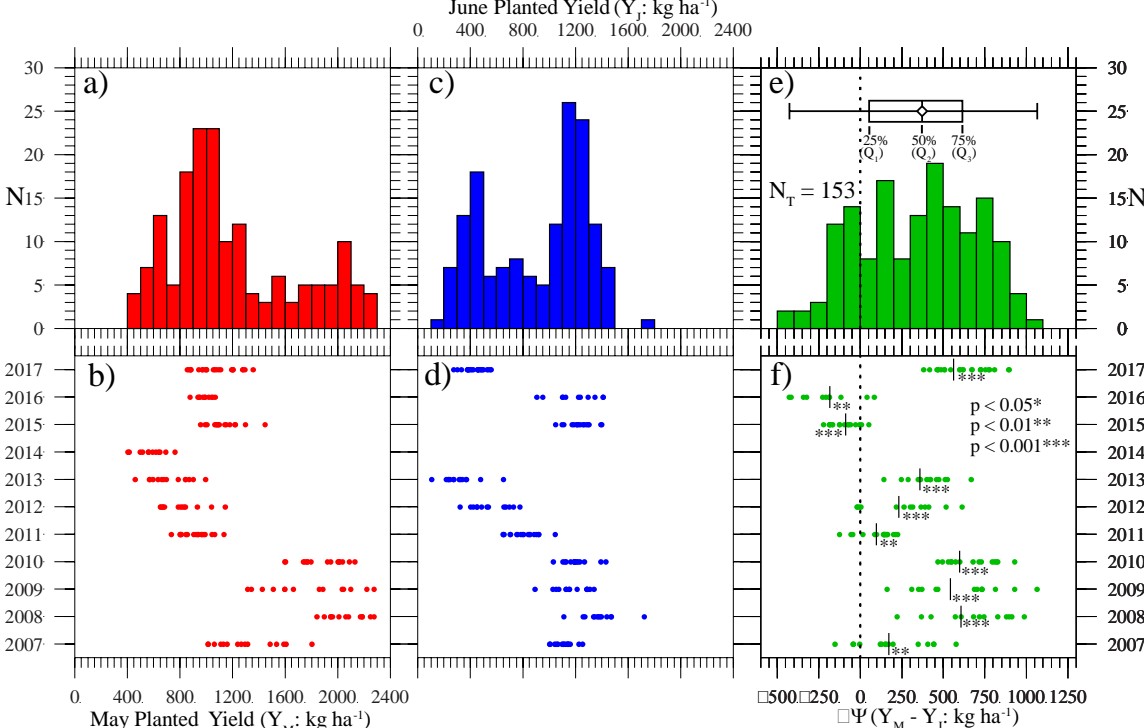

**Figure 3.** (**a**) Count histogram of May-planted lint yields from the 2007–2017 AgriLife irrigated variety trials. (**b**) May-planted lint yields by year. (**c**) As in (**a**) for June-planted yields. (**d**) As in (**c**) for June-planted yields. June-planted yields were not reported in 2014. (**e**) Count histogram of May-planted minus June-planted yield effects (ΔY) calculated from the same cultivar planted on May and June planting dates during each year. Bar plot marks the distribution's minimum and maximum, and 25th (Q1), 50th (Q2) and 75th (Q3) percentiles. (**f**) Yield effects plotted by year. Vertical lines mark sample means. Stars show mean effects significant at at least a 95% confidence level ($p < 0.05$) as determined by a one sample *t*-test.

Figure 3e's irrigated yield effects can be compared with the M19 simulated SHP planting date effects found in rainfed CROPGRO-Cotton model runs. Those simulations were repeated with daily weather inputs from Figure 1's 21 Mesonet stations over an 11-year period (2005–2010, 2012–2016) resulting in 231 simulated yields. As noted before, the median yield effect of planting on May 15 versus June 5 in the M19 rainfed simulations was 163.7 kg ha$^{-1}$, with positive yield effects found in 197 of 231 simulations (85.3%). The median yield effect in Figure 3e is 371.7 kg ha$^{-1}$ (330.8 lb ac$^{-1}$), with positive effects in 120 of 153 comparisons (78.4%). Thus the magnitude of the median irrigated field trial effect is about twice that of the simulated rainfed effect, while the incidence of positive yield effects are similar. Although caution should be used in comparing simulated yield outcomes based on one cultivar with observed yield outcomes from multiple cultivars, the higher magnitude of the AgriLife field trial effects is generally consistent with lower levels of water stress resulting in stronger GDD-related yield effects [1,5].

Figure 4 estimates the sensitivity of lint yields to the varying effects of GDD$_S$ accumulations beginning on May and June planting dates. The Figure 4a scatter point's Y-coordinates are the May-

and June-planted yields of Figure 3b,d, while the X-coordinates are the $GDD_S$ accumulated between the corresponding field experiment's planting and harvest dates in Table 1. Figure 4b scatter plots the Figure 3f $\Delta Y$ values vs. the corresponding difference in May and June-planted growing season $GDD_S$, i.e., Table 1's $\Delta GDD_S$ values. Figure 4c is the histogram of $\Delta Y / \Delta GDD_S$ (Y') ratios for each of Figure 4b's scatter points, which estimates the distribution of yield effect per growing degree day (kg ha$^{-1}$ °C$^{-1}$) in the 153 lint yield comparisons. The Figure 4c' bar plot marks the Y' histogram's quartiles, while Figure 4c'' is a bar plot that excludes the Y' ratios from 2015 and 2016.

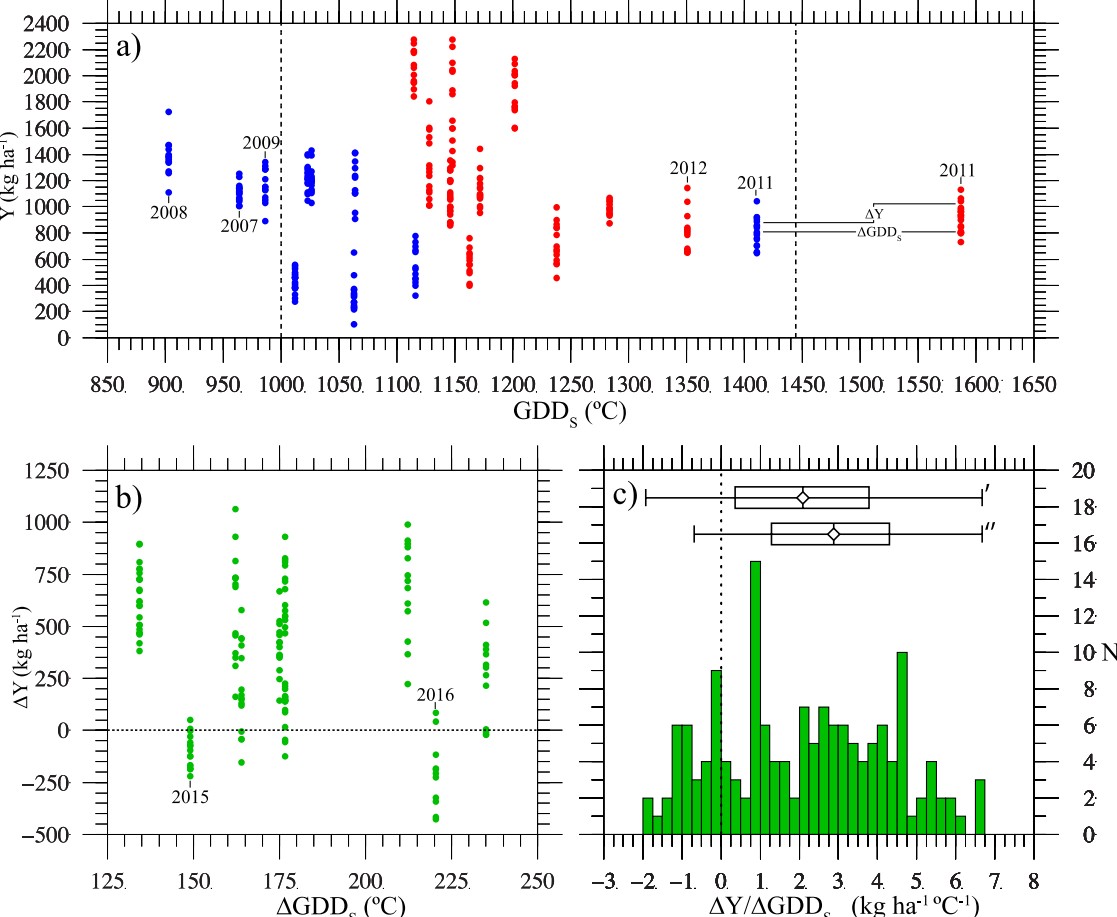

**Figure 4.** (**a**) Scatterplot showing May-planted (red) and June-planted (blue) lint yields on the Y-axis, and Equation (1) growing season cumulative growing degree days ($GDD_S$) on the X-axis. (**b**) Scatterplot showing May minus June yield effects ($\Delta Y$) for each cultivar-year on the Y-axis, and the corresponding difference in May minus June-planted growing degree days ($\Delta GDD_S$) on the X-axis. (**c**) Count histogram of the Y' ratios ($\Delta Y / \Delta GDD_S$) of the 153 scatter points in (**b**). Upper bar plot (**c'**) marks percentiles of all 153 ratio values. Lower bar plot (**c''**) marks percentiles of 111 values when 2015 and 2016 ratios are omitted.

Although seasonal GDD totals in the 1400.0–1500.0 °C range are commonly cited as necessary to bring a cotton crop to full maturity [1,31,35], in Figure 4a only 2011's June-planted or May-planted $GDD_S$ totals approach or exceed that range. Apart from that drought year [36], the highest May-planted $GDD_S$ total (1351.9 °C) occurred in 2012. In Figure 4a, May-planted $GDD_S$ totals in the 1100–1200 °C range produced yields as high as 2278.6 kg ha$^{-1}$ (2029 lb ac$^{-1}$). Based on Howell et al.'s [3] field experiments, Esparza et al. [35] proposed a 1000.0 °C (1800.0 °F) GDD threshold for profitable cotton production in the northern Texas and Oklahoma Panhandles and western Kansas. However, June planting dates in the 2007–2009 growing seasons accumulated less than that level, but produced

irrigated yields above 893.9 kg ha$^{-1}$ (747.5 lb ac$^{-1}$). This suggests, consistent with Esparza et al.'s conclusions, that with enough irrigation profitable cotton might be produced in the SHP with GDD$_S$ totals below 1000.0 °C, and well below that associated with full maturity.

In Figure 3f negative yield effects, i.e., those where June-planted yields exceeded May-planted yields, were common in 2015 and 2016. Figure 4b's $\Delta$Y vs. $\Delta$GDD$_S$ scatterplot might reveal whether these effects were associated with relatively small differences in growing season GDD$_S$. A weak GDD effect is possible in 2015, where the difference in May minus June-planted $\Delta$GDD$_S$ values (149.1 °C) was the second smallest of the 10 years. This may be due to the relatively small interval (22 days) between the May and June planting dates in the 2015 trials (Table 1). In terms of in-season rainfall, 2015 was also the wettest year of 2009–2017 (Table 3), which might mask weak GDD-related yield effects. But a weak GDD effect is less likely in 2016, where the difference in May-planted and June-planted seasonal GDD was the second largest ($\Delta$GDD$_S$ = 220.4 °C). The 2016 negative yield effects are most likely due to factors other than varying GDD accumulations over the course of the growing season. In particular, cool and wet conditions in May 2016 may have inhibited the early development of the May-planted crop, while the June-planted crop had more favorable emergence and seedling growth conditions.

The median yield effect per unit of GDD$_S$ in the Figure 4c' bar plot is 2.09 kg ha$^{-1}$ °C$^{-1}$, with a first quartile ($Q_1$) of 0.35 kg ha$^{-1}$ °C$^{-1}$ and a third quartile ($Q_3$) of 3.79 kg ha$^{-1}$ °C$^{-1}$. Given past field research that found lint yields to be positively correlated with seasonal GDD [1,5], more accurate estimates of yield sensitivity to GDD$_S$ might be made based on only those years with significant positive mean yield effects. When 2015 and 2016 ratios are omitted (Figure 4c''), the median yield effect increases to 2.88 kg ha$^{-1}$ °C$^{-1}$, with the central 50% of values bounded by 1.29 kg ha$^{-1}$ °C$^{-1}$ and 4.31 kg ha$^{-1}$ °C$^{-1}$.

Median yield effects found here with or without 2015 and 2016 Y' ratios are considerably higher than the 1.12 and 0.96 kg ha$^{-1}$ °C$^{-1}$ regression estimates of lint yield response to growing season GDD calculated by Peng et al. [1] and Wanjura et al. [5]. The higher values here may be due to improved genetics, a larger sampling of cultivars, exposure to a broader range of growing season conditions, or some combination of the three. With 2015 and 2016 Y' values omitted, the Figure 4c'' bar plot represents the outcomes of 8 years of field studies with 111 cultivars. Although the Wanjura et al. trials were conducted over 12 years (1988–1999), only 3 cultivars were planted. The Peng et al. trials were also conducted with 3 cultivars, but over 2 growing seasons (1980–1981). Even so, these trials all demonstrate that, to some degree, when SHP cotton is not water-limited more degree days produces more lint yield. As a result, management practices that increase GDD$_S$ totals will generally produce higher yields.

## 3.2. Planting Date Effects on Fiber Quality

Figure 5 is Figure 3's counterpart showing the effects of May versus June planting dates on fiber micronaire (M) during the AgriLife variety tests. The Figure 5a–d X-axes are divided into 5 regions associated with increasing lint values: discounted values with micronaire less than 3.5 or greater than 4.9, base values in the 3.5–3.6 and 4.3–4.9 ranges, and premium values in the 3.7–4.2 range. In Figure 5e' and subsequent bar plots outliers are identified based on the inter-quartile range (IQR = $Q_3 - Q_1$), i.e., those values below $Q_1 - 1.5 \times$ IQR or above $Q_3 + 1.5 \times$ IQR [37] (pp. 43–44).

Comparing the Figure 5a,c histograms shows that June planting dates resulted in a clearly higher incidence of low-discounted micronaire values. During 2007–2017, 67 of 162 June plantings (41.4%) resulted in micronaire <3.5, compared to 15 of 162 May plantings (9.3%). By contrast, high-discounted (>4.9) micronaire was rare in both the May (2.5%) and June (3.7%) plantings. The incidence of micronaire in the premium range was somewhat higher with earlier planting dates, with 62 (38.3%) in the May plantings and 46 (28.4%) in the June plantings. The higher incidence of low-discounted micronaire in the June plantings leads to a much lower incidence of June-planted base values (26.5%) relative to the May plantings (50.0%). In Figure 5d years with a clear tendency to June-planted micronaire less than 3.5 include 2008, 2012, 2013, and 2017. In Figure 5e's $\Delta$M histogram 73.5% (119 of 165) of the values are

greater than 0.0 with a median May-planted minus June-planted micronaire effect of 0.5. In Figure 5f eight of the 11 year's mean ΔM are positive, and in 7 of those years *t*-tests show that the sample means are inconsistent with a 0.0 population mean at a 99.9% confidence level or better (*p* < 0.001). However, two year's (2010, 2014) ΔM sample means are significantly negative at a 95% confidence level or better.

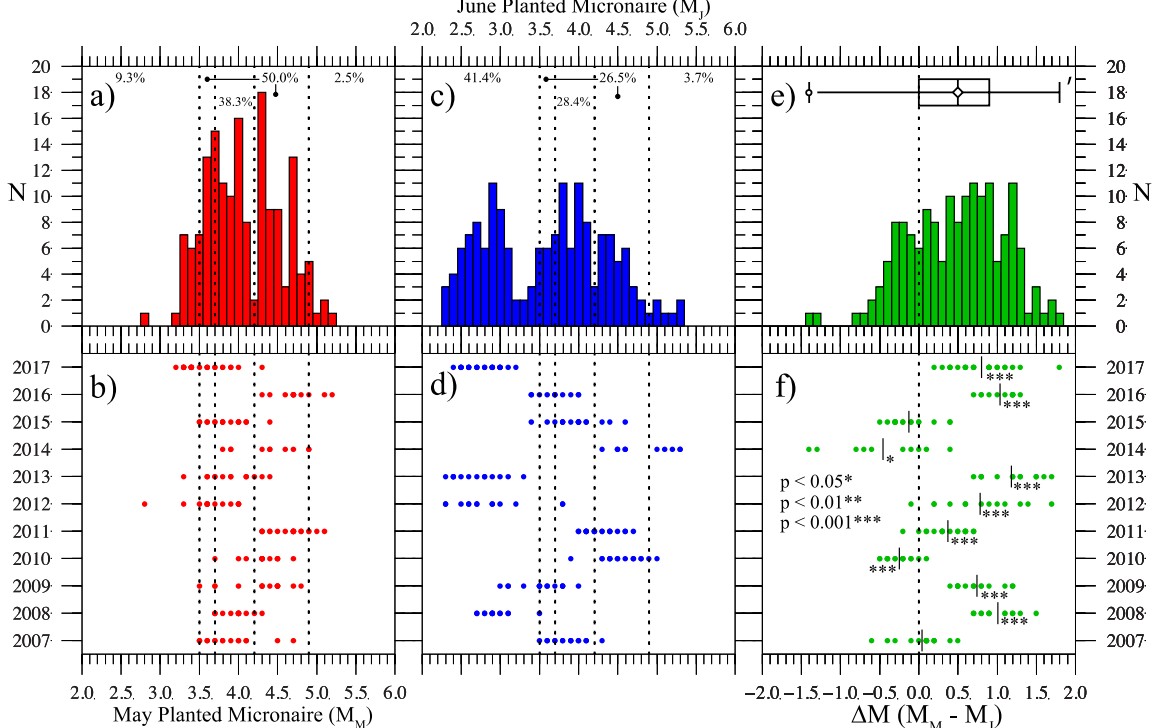

**Figure 5.** (**a**) Count histogram of May-planted micronaire values from the AgriLife irrigated variety trials. (**b**) May-planted micronaire by year. (**c**) As in (**a**) for June-planted micronaire. (**d**) As in (**c**) for June-planted micronaire. (**e**) Count histogram of May-planted minus June-planted micronaire effects (ΔM) calculated from the same cultivar planted on May and June planting dates during each year. Bar plot marks the distribution's minimum and maximum, and 25th, 50th and 75th percentiles. (**f**) Micronaire effects plotted by year. Vertical lines mark sample means. Stars show mean effects significant at at least a 95% confidence level (*p* < 0.05) as determined by a one sample *t*-test.

Following Figure 5e,f, Figure 6 shows the effects of May vs. June planting on fiber length (a), uniformity (b), strength (c), and elongation (d). During the HVI fiber analyses the length of lint samples is instrument-measured to the nearest hundredths of an inch. Uniformity is a measure of a sample's variability in fiber length, measured as the ratio of the average length of the entire sample to the average of the length of the sample's upper 50th percentile by weight. Strength is the force needed to break a sample measured in grams per tex. Elongation is the amount that a fiber will stretch before rupture, expressed as a percent of fiber length [38] (pp. 41–73). As in Figure 5f, the Figure 6a–d quality effects measure May-planted minus June-planted fiber properties, i.e., ΔL (length), ΔU (uniformity), ΔS (strength), and ΔE (elongation).

In Figure 6a two year's (2010, 2017) sample ΔL means are positively significant at a 99.9% confidence level while two (2011, 2016) are negatively significant at at least a 95% level. The remaining 7 means are insignificant (*p* > 0.05). A similar pattern of insignificance is found in Figure 6b's sample ΔU means, where four of the sample means differ significantly from 0.0 at a 95% confidence level (0.01 < *p* < 0.05), but 7 are insignificant. Figure 6c shows a significant positive mean fiber strength effect in five years, no significance in five years, and a significant negative effect in one year. Figure 6d shows mean elongation effects that are comparable to Figure 5f's mean micronaire effects, but negative. The negative ΔE means of 7 years are significant at at least a 95% confidence level, while two years are

significantly positive. The 14 positive ΔE values for 2009 are clear outliers, which may be associated with intermittent stands and smaller statistical samples in that year's trials [20]. Unlike the median effects for yield and the other four fiber quality variables, the median ΔE value of all 162 E comparisons is negative (−0.3). Thus, apart from 2009 and 2008, Figure 6d's relatively clear pattern of negative mean ΔE indicates that May planting reduces fiber elongation relative to June planting.

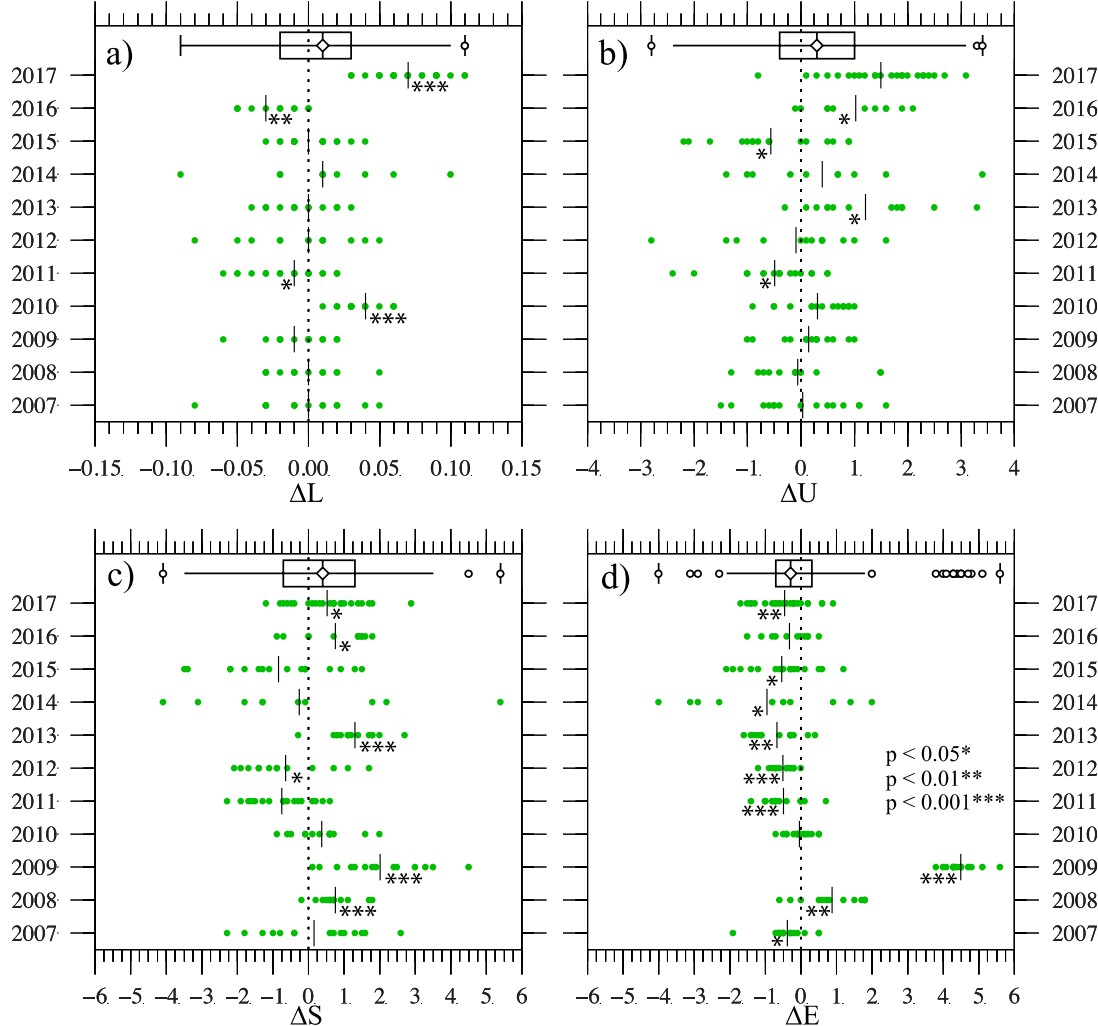

**Figure 6.** (**a**) Scatterplot of May-planted minus June-planted fiber length effects (ΔE) calculated from the same cultivar during each year. Bar plot marks 2007–2017 ΔE value's minimum and maximum, and 25th, 50th and 75th percentiles. (**b**) As in (**a**) for May-planted minus June-planted fiber uniformity effects (ΔU). (**c**) As in (**a**) for May-planted minus June-planted fiber strength effects (ΔS) (**d**) As in (**a**) for May-planted minus June-planted fiber elongation effects (ΔE).

Although mean elongation effects were significant in 7 of the 11 years, Figure 7's analyses focus on the effects of cumulative cool hours during boll formation (Equation (2)) on micronaire. Unlike elongation, micronaire variation affects lint value. Also, of the 5 quality variables evaluated, varying planting dates produced the highest incidence of positive or negative fiber quality effects in micronaire (73.5%), with 7 year's annual mean effects occurring at the highest confidence level (Figure 5f).

Figure 7 follows Figure 4 for the effects of CHRS on micronaire. The X-coordinates of the Figure 7a scatter points are the Equation (2) CHRS totals calculated based on each year's planting date, while the Y-coordinates are the May-planted and June-planted micronaire values. The Figure 7b X-coordinates plot Table 1's May-June ΔCHRS values for each year of 2007–2017, while the Y-coordinates plot those year's differences in May-planted and June-planted micronaire (ΔM) for each cultivar. All ΔCHR

values are negative, as each year's Table 1 May CHRS total is less than the June total. Figure 7c's histogram and bar plots plot the distribution of micronaire effect per cooling hour ($\Delta$M/$\Delta$CHRS) ratios for each point in Figure 7c, multiplied by 100.0 (M'). Thus Figure 7c estimates the distribution of the effects of 100 cooling hours on micronaire during a cotton crop's boll formation period.

In the Figure 7c histogram 73.5% of M' ratios are negative, which represent the effects of increased micronaire ($\Delta$M > 0) when cooling hours are reduced by 100 h ($\Delta$CHRS < 0). Reversing signs, negative M' can also be interpreted as the reduction in micronaire when cooling hours are increased.

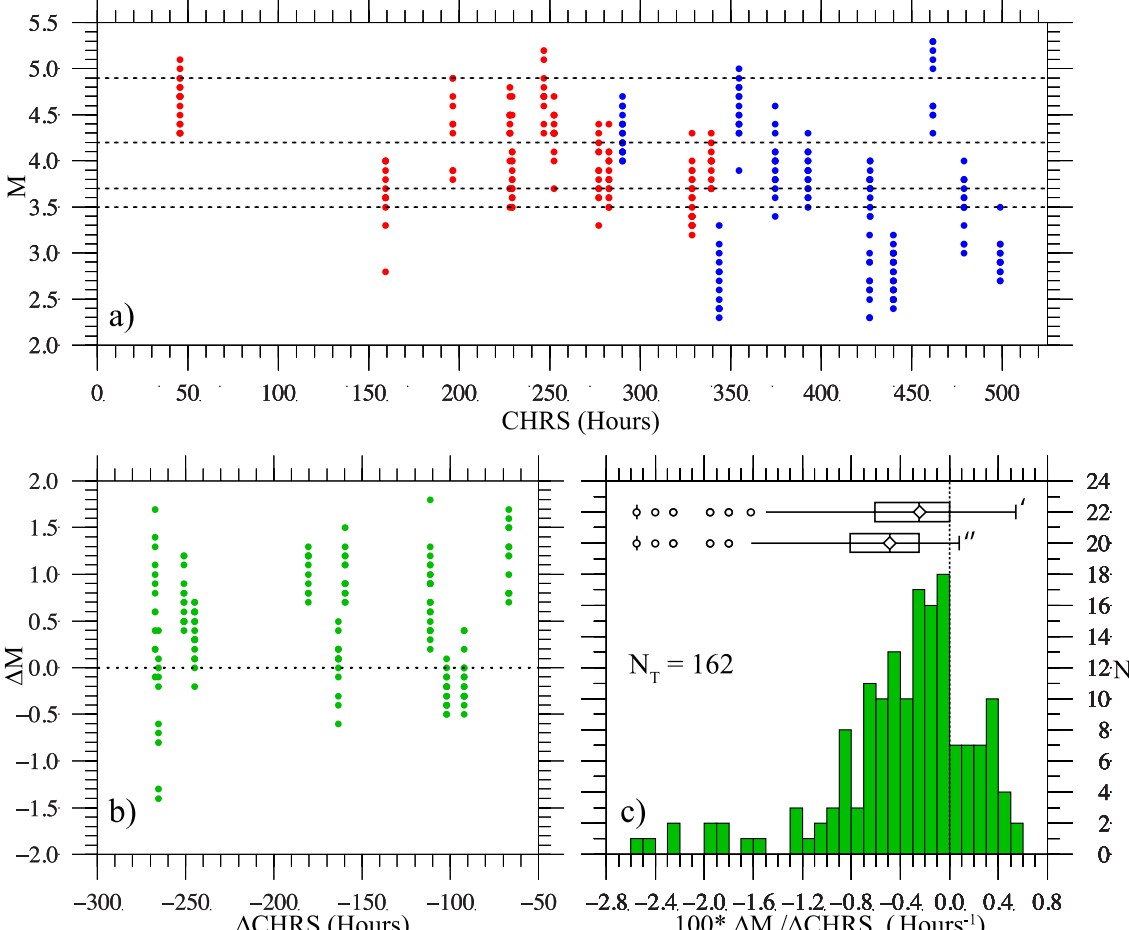

**Figure 7.** (**a**) Scatterplot showing May- (red) and June-planted (blue) micronaire on the Y-axis, and Equation (2) cooling hours (CHRS) on the X-axis. (**b**) Scatterplot showing May minus June micronaire effects ($\Delta$M) for each cultivar-year on the Y-axis, and the corresponding difference in May minus June-planted CHRS ($\Delta$CHRS) on the X-axis. (**c**) Count histogram of the M' micronaire sensitivity ratios (100.0 × $\Delta$M/$\Delta$CHRS) of the 162 scatter points in (**b**). Upper bar plot (**c'**) marks percentiles of all 162 M' values. Lower bar plot (**c''**) marks percentiles of 106 M' values for the years 2008, 2009, 2011–2013, 2016, and 2017.

The positive M' ratios reflect the 22.8% of cases in Figures 5f and 7b where May planting dates resulted in lower micronaire than that resulting from a June planting date ($\Delta$M < 0). The median effect of 100 cooling hours on micronaire during 2007–2017 was −0.245 (Figure 7c'), with the central 50% of values varying between −0.610 ($Q_1$) and 0.0 ($Q_3$). As the results of growth chamber and field experiments are consistent with changes in cool hours during boll formation leading to micronaire changes of opposite sign [7–9,32,33], a more representative distribution of field-based effects might result from only including M' ratios with positive $\Delta$M values. Alternatively, a distribution could be formed based on those years in Figure 5f with significantly positive $\Delta$M sample means (2008,

2009, 2011–2013, 2016, 2017). The distribution of those 106 M′ ratios in the Figure 7c″ bar plot has a −0.486 median, and a central 50% of values ranging between −0.809 and −0.245.

### 3.3. Environmental Controls on SHP Cotton Planting and Crop Development

The potential advantages of May SHP cotton planting include higher yields (Figure 3) and a lower incidence of micronaire in the low-discount range (Figure 5). But the region's altitude and short growing season can also restrict planting conditions, particularly those related to soil temperature at planting and air temperature in the days immediately after planting. Although the SHP is a relatively compact agricultural region by U.S. standards, a north-to-south temperature gradient introduces additional restrictions. Wanjura et al. [39] estimated that under field conditions, 103 h of seed-level temperatures above 18.0 °C (64.4 °F) were required for initial cotton emergence. Kerby et al.'s [40] multiple regression analyses found that seed quality and GDD accumulated in the 5 days after planting ($GDD_{5+}$) were the best predictors of emergence, accounting for 68.5% of variation in a 3-year field study. More recent guidelines in the southeastern U.S. [31] and the SHP [41] proposed by regional research and producer outreach ('extension') agencies recommend that soil temperature at ~10 cm (4 in.) should be above 18.3 °C (65.0 °F) over the 72-h period before planting, with warm conditions in the near-term forecast. Sansone et al. [41] (pp. 61–69) suggest maximum and minimum daily temperatures in the 5-day forecast resulting in degree-day accumulations of at least 13.9 °C (25.0 °F). Based on daily minimum and maximum 2.0 m air temperature data and hourly 10 cm soil temperatures ($ST_{10}$) from each of Figure 1's WTM stations, Figure 8a plots the first day-of-year when those $ST_{10}$ and $GDD_{5+}$ conditions were simultaneously met at each station during 2005–2017. That is, the first day-of-year ($pdoy_1$) in which a station's soil temperatures at 10 cm during the previous 3 days were all above 18.3 °C, and the $GDD_{5+}$ total for the following 5 days (Equation (3)) was greater than 13.9 °C.

$$GDD_{5+} \;=\; \sum_{d=pdoy+1}^{pdoy+5} Max(T_d - 15.6,\ 0.0) \qquad\qquad (3)$$

In Figure 8a WTM station latitudes define the scatter point's Y-coordinates, while X-coordinates mark the first day of each year when the $pdoy_1$ condition was met at each station. The $pdoy_1$ dates generally occur later as latitude increases, with dates ranging between April 2 and June 3 in the southern (black) scatter points, and April 14 and June 5 in the northern (gray) scatter points. The median date for the southern stations is April 27 (doy 117), while the median for the northern stations is May 14 (doy 134). If the southern and northern $pdoy_1$ percentiles are considered climatologically representative, i.e., estimates of probable outcomes under current SHP climate conditions, there is a 75% probability that the Sansone et al. planting condition will occur after April 19 in the south, and after May 8 in the north.

Figure 8b plots percentiles of $GDD_S$ (Equation (1)) for 18 planting dates ranging from March 1 to June 28 at 7-day intervals. The gray bar plots are formed from the 143 northern station $GDD_S$ values summed between each planting date and October 15 during 2005-2017, while the black bar plots show the corresponding percentiles for the 130 southern station $GDD_S$ values. In addition, the Figure 8b black and gray diamond tokens mark the medians of southern and northern $GDD_{5+}$ for each planting date. Consistent with a north to south temperature gradient, southern $GDD_S$ percentiles exceed the northern percentiles for the same planting date. This latitudinal effect's magnitude might be estimated from the March 1 percentiles, which give estimates of the distributions of 'potential $GDD_S$', i.e., maximized totals that do not exclude degree days accumulated during the April–June period because of delayed planting. The March 1 median $GDD_S$ values for the southern and northern WTM stations are 1370.9 and 1204.8 °C respectively. As spring degree days begin to accumulate, median $GDD_S$ for subsequent March and April planting dates decrease slightly from these potential values, with greater decreases as $GDD_{5+}$ totals increase throughout May and June. With June 28 planting dates median southern $GDD_S$ drops to 895.8 °C, while median $GDD_S$ for the northern stations drops to 810.2 °C.

The Figure 8c bar plots show the northern and southern CHRS percentiles during a boll formation period 80 to 110 days after each of the 18 planting dates (Equation (2)). Again, consistent with generally cooler northern conditions, the northern percentiles exceed each date's corresponding southern percentile. Over both northern and southern stations median CHRS decrease throughout March planting dates, are relatively low during April, then steadily increase after May 10. Given the Equation (2) lag period, the CHRS minima over the four April planting dates reflects a maxima in summer nighttime temperatures between June 29 (doy 180) and August 14 (doy 226). The lowest median values in the north (205.0 h) and the south (151.9 h) occur with April 19 (doy 109) planting dates. Assuming a boll formation period 80 to 110 days after planting, that date would result in boll formation during the SHP summer growing season's climatologically warmest 31-day period, which occurs between July 8 and August 7th. By comparison, May 24 planting would result in boll formation during the cooler August 12 to September 11 period, with median northern (southern) CHRS totals of 301.0 (273.0) hours.

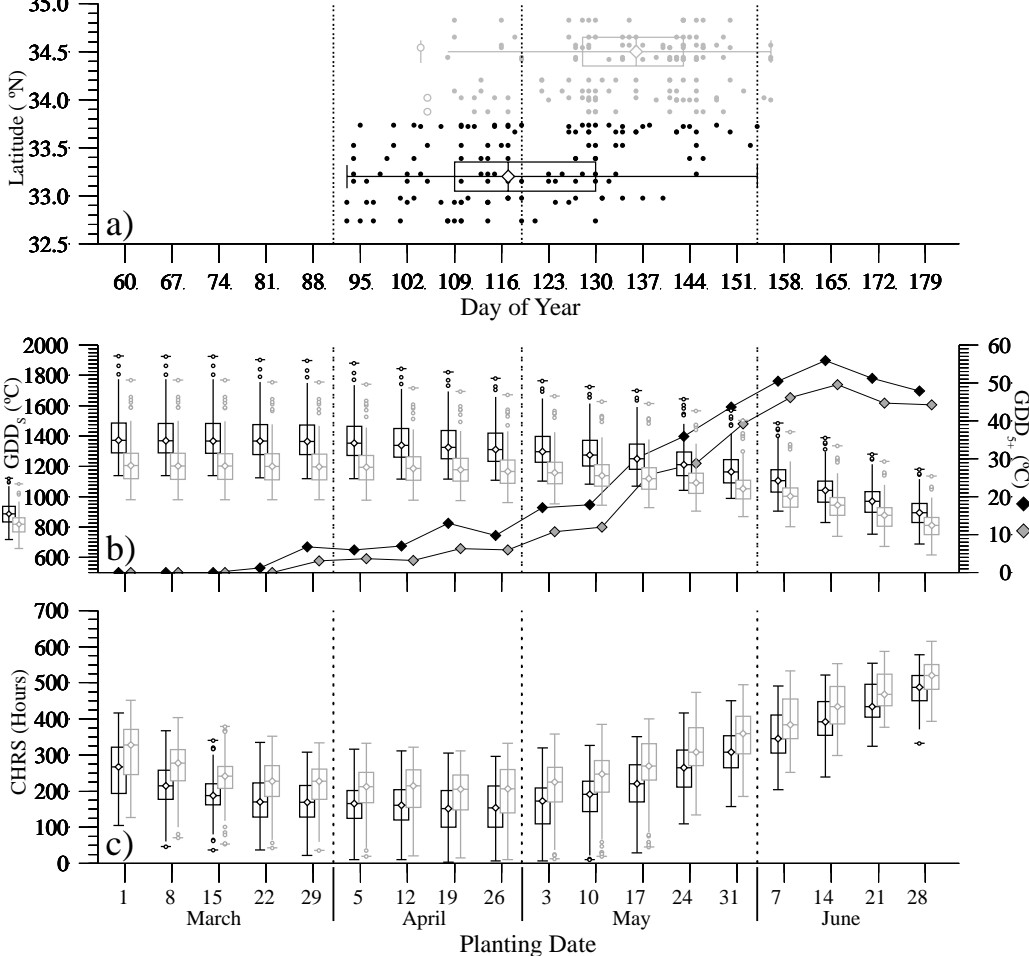

**Figure 8.** (**a**) Scatterplots of Figure 1 station latitudes (X-axis) vs. first day of year (pdoy$_f$) Sansone et al. [41] soil temperature and GDD$_{5+}$ planting conditions were met during each year of 2005-2017 (Y-axis). Gray (black) points show scatter of Figure 1's 11(10) northernmost (southernmost) stations. Gray (black) bar plots mark percentiles of northern (southern) pdoy$_f$ values. (**b**) Gray (black) bar plots showing the percentiles of GDD$_S$ (Equation (1)) calculated between 18 planting dates and Oct. 15 during 2005–2017 over Figure 1's northern (southern) WTM stations. Gray (black) diamonds show medians of northern (southern) growing degree days accumulated in the 5 day period after each planting date (GDD$_{5+}$) (**c**) Gray (black) bar plots showing the percentiles of cooling hours (CHRS: Equation (2)) accumulated 80 to 110 days after 18 planting dates during 2005–2017 over Figure 1's northern (southern) WTM stations.

## 4. Discussion

This analysis of AgriLife irrigated variety trial data and regional (WTM) temperature data shows that additional growing degree days resulting from May planting can substantially increase lint yields relative to June planting. May planting may also decrease exposure to cooling hours during a crop's boll formation period and reduce the likelihood of discounted low fiber micronaire. Under typical planting practices SHP producers normally plant before June 5 or June 10 to qualify for federal crop insurance. Planting after those dates is usually re-planting after hail damage. During 2013–2017 the average percentage of cotton acres planted in Texas by May 7 to May 13 was 23% [42], thus most planting currently takes place between mid-May and early June. As a result, most of the AgriLife June planting outcomes used here to estimate planting date effects are not generally representative of SHP cotton production. But, given the apparent advantages of earlier planting, what planting dates might be optimal under the SHP region's current climate conditions? In both southern and northern growing areas, increasing $GDD_{5+}$ accumulation begins in early to mid-May, with planting dates after that time resulting in lower seasonal $GDD_S$ totals (Figure 8b). Assuming a boll formation period 80 to 110 days after planting, April and early-May planting dates minimize a SHP cotton crop's later exposure to CHRS (Figure 8c). As a result, late-April to early-May planting dates would simultaneously maximize $GDD_S$ and minimize CHRS, with potentially related effects of increased yield and micronaire.

However, the SHP spring planting environment can limit when planting can occur. In rainfed production planting can be delayed as producers wait for planting rains to increase soil moisture at seed depth. Temperature conditions below and above ground can also restrict planting, with greater restrictions in more northern production areas. Extension guidelines [41] propose that planting depth soil temperature should be above 18.3 °C (65.0 °F) in the 72 h before planting, and that forecast GDD in the 5 days after planting ($GDD_{5+}$) be above 13.9 °C (25.0 °F). Analysis here based on WTM air and soil temperature data estimates that, in the current SHP climate, there is a 75% probability that these planting conditions would first occur after April 19 south of Lubbock, and after May 8 north of Lubbock (Figure 8a). But northern planting dates after May 8, which might be considered to climatologically occur in 3 of every 4 years, do not take advantage of available spring degree-days in an already degree-day limited environment. Later mid-May to early-June planting dates also increase the chances of boll formation taking place under cooler conditions (Figure 8c).

Although most SHP cotton acres are currently planted after mid-May, late-April to early-May planting dates might improve cotton production and profitability in more northern areas. While low soil temperatures may restrict germination during this period, Sansone et al. [41] suggest that temperatures above 15.6 °C (60.0 °F) in the 3 days before planting may be acceptable with a favorable $GDD_{5+}$ forecast. Planting high vigor seed with higher germination rates in cooler soils [43] may provide more consistent stands under these conditions. As dry spring soils might also inhibit germination, this early planting strategy may be most useful to producers with access to irrigation. Early planting may also introduce a trade-off between improved micronaire and fiber that ruptures more easily, as May planting was found here to both increase micronaire and decrease elongation. But given increased irrigated yields (Figure 3e,f) and a lower risk of low-discounted micronaire with earlier planting dates (Figure 5a–d), the profit benefits of earlier planting may outweigh the risks.

Developing upland cotton genetics that are optimized to the SHP environment may be the key to reducing the risks associated with earlier planting. The results here suggest a number of traits that would make cotton cultivars better suited to late April or early May planting, or more generally suited to SHP production. Plant breeding efforts to tailor cotton genetics to cooler SHP planting conditions might be directed to emphasizing seed vigor and cold germination. Until these traits are developed, other breeding efforts might be directed to developing cultivars with maturity traits specifically adapted to SHP planting and growing conditions. Although boll formation was assumed here to occur 80–110 days after planting, cultivars adapted to SHP conditions should, regardless of planting date, possess maturity traits that allow for complete boll formation before cool hours begin accumulating in late August and early September. If planting dates resulting in August-September

boll formation periods are unavoidable, cultivars bred to be less sensitive to cool conditions during boll formation might be planted, e.g., cross-bred strains developed from low M' cultivars. Generally, a cultivar adapted to the SHP environment should be capable of high yields under degree day-limited growing conditions, i.e., should possess high $GDD_S$ sensitivity and high Y'. The Y' and M' ratios calculated here may be useful yardsticks in identifying cultivars that meet these requirements. Given the corresponding low M' and high Y' cultivars, which are listed in the LBKFiberQual2007–2017.xlsx and LBKYield2007–2017.xlsx supplemental files, a breeding and heritability study similar to that of Ulloa [44] may be an important first step in developing genetics tailored to the climate of the SHP and other cool growing environments.

**Supplementary Materials:** The following are available online at http://www.mdpi.com/2077-0472/9/4/82/s1, supplementary file 1: LBKYield2007–2017.xlsx, supplementary file 2: LBKFiberQual2007–2017.xlsx.

**Author Contributions:** J.D. led the collection, organization and reporting of lint yield and fiber quality data. S.M. and M.U. designed the statistical analysis. S.M. wrote the first draft and analyzed the data. M.U. and J.D. reviewed and edited the final draft. All authors assisted in writing and improving the paper.

**Funding:** This study was funded by USDA-ARS Projects: 3096-13000-009-00-D (SM) and 3096-21000-019-00-D (MU), and USDA CSREES Hatch Project TEX 09297 (JD).

**Acknowledgments:** All figures were produced using Generic Mapping Tools [45]. The mention of trade or manufacturer names is made for information only and does not imply an endorsement, recommendation, or exclusion by the USDA-Agricultural Research Service. This research did not receive any specific grant from funding agencies in the commercial or not-for-profit sectors. The USDA is an equal opportunity provider and employer.

**Conflicts of Interest:** The authors declare no conflict of interest. Funding sponsors did not play a role in study design, data collection, interpretation, or analysis, or the decision to publish results.

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
