# Peer review of "Planting Date Effects on Cotton Lint Yield and Fiber Quality in the U.S. Southern High Plains"

_agriculture, doi:10.3390/agriculture9040082_

Reviewer 1 Report

In this paper, the planting date effects on cotton lint yield and fiber quality in the U.S. Southern High Plains were discussed in detail. As a whole, the review is quite comprehensive and well written.

Introduction: Please include the significance of the review

Table 2: please provide the standard deviation value in the table

Page 15: Please include a graphical representation (overall representation) of the effect of planting date on the lint yield and fibre quality.

Discussion (section 4): Please include the application of the study

Author Response

In this paper, the planting date effects on cotton lint yield and fiber quality in the U.S. Southern High Plains were discussed in detail. As a whole, the review is quite comprehensive and well written. 

Rev1-1: Introduction: Please include the significance of the review

Reply:We assume this refers to ‘define the purpose of the work and its significance’ in the authors instructions.On lines 107-110 of the current draft we have added the following 2 sentences (and two additional references) to make the connection between our work and the more general ‘genetics X environment X management’ movement in agricultural research. 

‘In agricultural research there is a growing recognition that producing crops successfully in changing or marginal environments may require an understanding of the interactions between genetics, environment, and management [28,29]. Based on this understanding, management and genetics might be adapted to a region’s environment.’

Rev1-2: Table 2: please provide the standard deviation value in the table

Reply:Table 2 lists the number of cultivars planted in each year’s field trial, but we are not sure  why the inter-annual variability of the number of cultivars planted each year in the trials is an important statistic in the analysis.  Could the reviewer explain why she/he thinks this would be a useful addition to the table? The standard deviations for the 2007-2017 cultivar counts are included in the table below.

Rev1-3: Page 15: Please include a graphical representation (overall representation) of the effect of planting date on the lint yield and fibre quality.

Reply:We’d like to avoid adding additional figures. There are already 2 figures in the paper that show the effect of planting date on lint yield (Fig. 3e,f ) and 6 figures that show fiber quality effects (Fig. 5 e,f, Fig. 6a-d).

Rev1-4: Discussion (section 4): Please include the application of the study.

Reply: The specific application of the study is to propose management practices that are adapted to the largest cotton production region in the U.S. For a more general application, in the current Introduction on lines 107-110 we’ve tried to make a general case for our work as an example of  ‘genetics X environment X management’ research. To further generalize our results, we’ve changed the last sentence of Section 4 (on lines 547-550) to suggest that improved cultivars that may result from related future research will also be adapted to other high altitude regions with short growing seasons.

‘Given the corresponding  low M’ and high Y’ cultivars, which are listed in the LBKFiberQual2007-2017.xlsx and LBKYield2007-2017.xlsx supplemental files, abreeding and heritability study similar to that of Ulloa [44]may be an important first step in efforts to develop genetics tailored to the climate of the SHP and other high altitude growing regions.’

Reviewer 2 Report

Review of Mauget et al., Planting Date Effects on Cotton Lint Yield and Fiber Quality in the U.S. Southern High Plains, Agriculture

General overview: The authors conduct multiple analyses to determine how planting date contributes to cotton quality, noting that genetic improvement and selective breeding could be targeted to improve outcomes for growers in the US Southern High Plains.  Conclusions are based on substantial analysis of long-term data, and data supports conclusions.

General comments: For an international journal, my only comment is that this is may appeal only to a very narrow niche.  I couldn’t speak for the Journal Editor on how important this may or may not be for the Journal. As a methods papers, or a case study of sorts, the reach is broader.  It’s certainly an interesting use of the available data to optimize cotton production in the area of focus.  This would not be the basis for any decisions on publication acceptability on my end.

It could be instructive for non-cotton audiences if just a few sentences could be included in the Introduction to summarize the factors that go into “quality” assessment of cotton. The paragraph from ln 54-78 discusses various aspects of quality (e.g., micronaire, fiber length, fiber strength), but as presented, it’s not assembled into one concise statement. This could be as simple as “high quality cotton is characterized as having ……”  This would frame the information presented on previous research.

The presentation format for Results tends to bury the lead, and is challenging to read. While I understand that the results are graphics heavy, the Results section reads more like a series of extended figure captions (for example, ln 317-329). It might shorten the (lengthy) paper, while also increasing the readability, if the format were altered to refer to the figures when making an observation, rather than describing each figure in the text.

Ln283-284 In these lines, the conclusion is that profitability is propped up on the use and availability of irrigation. Is there a point where irrigation becomes too costly to maintain profitability? With dwindling aquifers worldwide, is there a near-term risk irrigation can’t meet the need? If either of these situations is looming just beyond the horizon, it might be worth considering how that plays into your overall conclusions.

Extension is mentioned at several points in the paper (ln 425, 508). The role and purpose of Extension may not be obvious to non-US audiences, thus I’m not confident that simply calling it “extension,” particularly with a lower-case e, will adequately indicate what you mean.

Suggested edits: 

For clarity, a few commas/hyphens could be added, and a few tricky verbs.

Ln21 “In 7 of 11 years, May…”

Ln 42 “…elevation of SHP cotton yields, Maguet et al….”

Ln63 replace “was” with “were” as the sentence breaks down to “accumulations were associated”

Ln81 “In early November 2017, 47% of the…”

Ln115 “In the following, Section 2 will describe…”

Ln127 “…accumulated between planting and harvest, and exposure to …”

Ln151 “…temperatures in the 5-minute Abernathy data,….”

Ln166 “….WTM stations was used…” as this is all past tense

Ln184 “In 2014, early-season cool temperatures….”

Ln218 replace “are” with “is” as the sentence breaks down to “code is included”

Ln231 “As in Fig. 2, May data….”

Ln277 “In Fig. 4a, May planted GDDs totals” (note here also that the period is missing after Fig)

Ln290 “In terms of in-season rainfall, 2015 was…”

Ln339 “During 2007-2017, 67 of 162 June…”

Ln421 “…field conditions, 103 hours of seed-level….”

Ln506 “In rainfed production, planting can be delayed…”

Ln509 “…propose that planting-depth soil temperature…”

Author Response

Review of Mauget et al., Planting Date Effects on Cotton Lint Yield and Fiber Quality in the U.S. Southern High Plains, Agriculture

General overview: The authors conduct multiple analyses to determine how planting date contributes to cotton quality, noting that genetic improvement and selective breeding could be targeted to improve outcomes for growers in the US Southern High Plains.  Conclusions are based on substantial analysis of long-term data, and data supports conclusions.

General comments: For an international journal, my only comment is that this is may appeal only to a very narrow niche.  I couldn’t speak for the Journal Editor on how important this may or may not be for the Journal. As a methods papers, or a case study of sorts, the reach is broader.  It’s certainly an interesting use of the available data to optimize cotton production in the area of focus.  This would not be the basis for any decisions on publication acceptability on my end.

Rev2-1: It could be instructive for non-cotton audiences if just a few sentences could be included in the Introduction to summarize the factors that go into “quality” assessment of cotton. The paragraph from ln 54-78 discusses various aspects of quality (e.g., micronaire, fiber length, fiber strength), but as presented, it’s not assembled into one concise statement. This could be as simple as “high quality cotton is characterized as having ……”  This would frame the information presented on previous research.

Reply:On lines 55 and 56 of the current draft we’ve added the following sentence:

High quality cotton fiber is uniformly strong, fine, long, and white, and is easily ginned and processed.

Rev2-2: The presentation format for Results tends to bury the lead, and is challenging to read. While I understand that the results are graphics heavy, the Results section reads more like a series of extended figure captions (for example, ln 317-329). It might shorten the (lengthy) paper, while also increasing the readability, if the format were altered to refer to the figures when making an observation, rather than describing each figure in the text.

Reply:  Our problem with referring to a figure while making an observation is that readers may be confused without a clear idea of what the figure communicates and how it is organized. This may a be a particular problem with multi-pane figures like Figs 3 and 5. Whether the figure is described before making an observation about the figure, or while the observation is being made, the figure has to be described in the text at some point. To make sure readers understand what we’re discussing when we describe a figure-based result, our approach is 1) describe the figure, then, 2) discuss the figure. This approach also leads the reader to look at the figure before its results are described in the text. Even so, in the discussion of Fig. 4 we’ve shortened the figure description on lines 269-278, omitted some redundant sentences in the paragraph describing Fig. 5 on page 9 (lines 330-335), and edited out sentences describing Figs. 8a and b  on page 13.

Rev2-3: Ln283-284 In these lines, the conclusion is that profitability is propped up on the use and availability of irrigation. Is there a point where irrigation becomes too costly to maintain profitability? With dwindling aquifers worldwide, is there a near-term risk irrigation can’t meet the need? If either of these situations is looming just beyond the horizon, it might be worth considering how that plays into your overall conclusions.

Reply:We assume this is about lines 280-282 in the version we downloaded: 

“This suggests, consistent with Esparza et al.’s conclusions, that with enough irrigation profitable cotton might be produced in the SHP with GDDStotals below 1000.0 °C, and well below that associated with full maturity.”

The point where irrigation becomes too costly to maintain profitabilitydepends upon pumping costs (and other production costs) and yield revenues. The last is mainly a function of commodity prices. If lint prices are low enough even highly irrigated crops might not be profitable. The point we were trying there is that even though our analysis was centered on temperature effects, water is the leading factor in driving yields. But if water is not a limiting factor, yields will be proportional to accumulated GDD over the course of the growing season. 

Rev2-4: Extension is mentioned at several points in the paper (ln 425, 508). The role and purpose of Extension may not be obvious to non-US audiences, thus I’m not confident that simply calling it “extension,” particularly with a lower-case e, will adequately indicate what you mean.

Reply:On lines 432-434 of the current draft we’ve changed a sentence to give readers a definition of U.S. agricultural extension.

“More recent guidelines in the southeastern U.S. [29]  and the SHP  [39]proposed by regional research and producer outreach (‘extension’)agencies....”

Suggested edits: 

For clarity, a few commas/hyphens could be added, and a few tricky verbs.

Ln21 “In 7 of 11 years, May…” (Not done, introductory phrase is short enough that we feel a comma isn’t necessary)

Ln 42 “…elevation of SHP cotton yields, Maguet et al….” (Done)

Ln63 replace “was” with “were” as the sentence breaks down to “accumulations were associated” (Done)

Ln81 “In early November 2017, 47% of the…” (Done)

Ln115 “In the following, Section 2 will describe…” (Done)

Ln127 “…accumulated between planting and harvest, and exposure to …” (Done)

Ln151 “…temperatures in the 5-minute Abernathy data,….” (dash inserted)

Ln166 “….WTM stations was used…” as this is all past tense (Done)

Ln184 “In 2014, early-season cool temperatures….” (Done)

Ln218 replace “are” with “is” as the sentence breaks down to “code is included” (Done)

Ln231 “As in Fig. 2, May data….” (Done)

Ln277 “In Fig. 4a, May planted GDDs totals”. Note here also that the period is missing after Fig. (Both Corrected)

Ln290 “In terms of in-season rainfall, 2015 was…” (Done)

Ln339 “During 2007-2017, 67 of 162 June…” (Not done, introductory phrase is short enough that a comma doesn’t seem necessary)

Ln421 “…field conditions, 103 hours of seed-level….” (Done)

Ln506 “In rainfed production, planting can be delayed…” (Not done, introductory phrase is short enough that a comma isn’t necessary)

Ln509 “…propose that planting-depth soil temperature…”(Done)
